# Chemical Variation and Environmental Influence on Essential Oil of *Cinnamomum camphora*

**DOI:** 10.3390/molecules28030973

**Published:** 2023-01-18

**Authors:** Ting Zhang, Yongjie Zheng, Chao Fu, Haikuan Yang, Xinliang Liu, Fengying Qiu, Xindong Wang, Zongde Wang

**Affiliations:** 1College of Forestry, Jiangxi Agricultural University, East China Woody Fragrance and Flavor Engineering Research Center of National Forestry and Grassland Administration, Nanchang 330045, China; 2Camphor Engineering and Technology Research Center of National Forestry and Grassland Administration, Jiangxi Academy of Forestry, Nanchang 330032, China

**Keywords:** *Cinnamomum camphora*, essential oil, oil yield, chemotypes, environmental factors

## Abstract

*Cinnamomum camphora* is a traditional aromatic plant used to produce linalool and borneol flavors in southern China; however, its leaves also contain many other unutilized essential oils. Herein, we report geographic relationships for the yield and compositional diversity of *C. camphora* essential oils. The essential oils of 974 individual trees from 35 populations in 13 provinces were extracted by hydrodistillation and analyzed qualitatively and quantitatively by gas chromatography-mass spectrometry and gas chromatography-flame ionization detection, respectively. Oil yields ranged from 0.01% to 3.46%, with a significantly positive correlation with latitude and a significantly negative correlation with longitude. In total, 41 compounds were identified, including 15 monoterpenoids, 24 sesquiterpenoids, and two phenylpropanoids. Essential oil compositions varied significantly among individuals and could be categorized into various chemotypes. The six main chemotypes were eucalyptol, nerolidol, camphor, linalool, selina, and mixed types. The other 17 individual plants were chemotypically rare and exhibited high levels of methyl isoeugenol, methyl eugenol, δ-selinene, or borneol. Eucalyptol-type plants had the highest average oil yield of 1.64%, followed in decreasing order by linalool-, camphor-, mixed-, selina-, and nerolidol-type plants. In addition, the five main compounds exhibited a clear geographic gradient. Eucalyptol and linalool showed a significantly positive correlation with latitude, while selina-6-en-4-ol was significantly and negatively correlated with latitude. trans-Nerolidol and selina-6-en-4-ol showed significantly positive correlations with longitude, whereas camphor was significantly and negatively correlated with longitude. Canonical correspondence analysis indicated that environmental factors could strong effect the oil yield and essential oil profile of *C. camphora*.

## 1. Introduction

*Cinnamomum camphora*, commonly known as the camphor tree, is a large evergreen tree in the Lauraceae family, which is native to China, Japan, and adjoining regions of South-East Asia [1]. It is an important commercial, timber, ornamental, and ecological tree species that is widely planted in the area south of the Yangtze River Basin in China [2]. Because of their beautiful shape, *C. camphora* trees have been introduced or cultivated as ornamental and street trees in many countries, including Australia, Holland, the Philippines, the USA, Cuba, and Sri Lanka [1]. In addition, many terpenoids are derived from their roots, trunks, branches, and leaves; thus, *C. camphora* is an important source of natural camphor and other fragrances and spices.

Linalool, camphor, and borneol derived from *C. camphora* are the most traded and consumed components of its essential oil. Among these compounds, borneol has various pharmaceutical functions in traditional Chinese medicine [3,4,5]. *C. camphora* is cultivated as an industrial crop in southern China, and its aerial parts are harvested in autumn to produce linalool or borneol. The camphor essential oil industry has become a characteristic industry, generating more than 10 billion Chinese yuan per annum, and the planting area in Jiangxi Province exceeds 10 thousand ha. *C. camphora* essential oil is compositionally diverse, containing nearly 100 detected compounds, including terpenes (such as myrcene, fenchol, and *α*-terpineol) and phenylpropenes (such as eugenol and safrole) [6,7,8].

Plant essential oils contain somewhat volatile secondary metabolites produced in response to biological and abiotic stressors. Various environmental factors affect the biosynthesis and accumulation of plant essential oils; thus, the content and composition of essential oils vary within the same species depending on location [9,10]. Chemotyping is an efficient way to describe individuals with similar essential oil compounds. *C. camphora* has been classified into different chemotypes based on compositional differences of the leaf essential oil, and the chemotypes vary in different regions [11,12,13]. Five chemotypes of *C. camphora*, named according to their dominant essential oil, are common in Jiangxi Province, including camphor, linalool, eucalyptol, isonerolidol, and borneol types [11]. The safrole and mixed types, featuring balanced oil composition, and three other chemotypes (linalool, eucalyptol, and camphor types) have been identified in Fujian Province [12]. Liu et al. [13] divided *C. camphora* from Guangxi Province into eucalyptol, linalool, camphor, terpene-terpineol (rich in *α*-pinene, sabinene, and 4-terpineol), and nerolidol types.

While the characteristics of camphor tree essential oils have been thoroughly studied, the large-scale essential oil diversity of camphor trees has been relatively overlooked. Chemotypic diversity may be related to genetic, physiological, and environmental factors, including climatic, topographic, hydrological, and edaphic factors [14,15,16,17]. The objective of this study was to characterize the chemotypic diversity of the major native camphor tree populations in China. A total of 974 individuals were studied from 35 populations ranging geographically from Sichuan to Zhejiang and from Shanxi to Guangdong. This study provides insight into the intra- and inter-population diversity of essential oils derived from camphor trees in China and a basis for the utilization and conservation of camphor tree resources.

## 2. Results

### 2.1. Variability in Essential Oil Yield

The leaf essential oils of 974 *C. camphora* individuals from 35 populations in 13 provinces were extracted and analyzed. A distribution interval of 0.2% was used to draw the frequency distribution map (Figure 1). The average essential oil yield of all individuals was 0.95%, and different individuals presented yields in the range of 0.01–3.46%. As shown in Figure 1, 176 individuals had essential oil yields less than 0.2%, and 181 individuals had essential oil yields between 0.2% to 0.4%. Nearly half of the individuals had relatively low essential oil yields (<0.6%), while a few individuals (53) had yields of >2.20%, accounting for 5.44% of the total samples. Additionally, several individuals with particularly high essential oil yields of >3.0%, including HB-ES-10 (3.46%), AH-QM-20 (3.28%), JX-AF-11 (3.24%), and JX-AF-38 (3.03%), were potentially excellent individuals for commercial development.

There was a significant difference among the essential oil yields of the different populations (*p* < 0.01). Camphor trees from Qimen had the highest mean essential oil yield (2.01%), followed by those from the Xinhua, Hanzhong, and Enshi populations, with mean oil yields of 1.70%, 1.54%, and 1.51%, respectively (Table 1). These four populations had high oil yields (>1.50%). Two populations from Guangdong (GD-SG and GD-ZJ), two from Fujian (FJ-FZ and FJ-HA), one from Jiangsu (JS-ZJ), one from Sichuan (SC-LX), and one from Zhejiang (ZJ-YY) exhibited low essential oil yields. Pearson’s correlation coefficient analysis revealed that essential oil yield was positively correlated with latitude. In other words, trees growing at high latitudes had higher oil yields than those growing at low latitudes. In contrast, essential oil yield was negatively correlated with longitude; higher oil yields were detected in camphor trees in western China (Table 2).

### 2.2. Essential Oil Profiles

A total of 41 compounds, including 15 monoterpenoids, 24 sesquiterpenoids, and two phenylpropanoids, were identified in the leaf essential oil of the camphor tree (Table 3). Interestingly, the vast majority of individuals in natural populations had distinct single principal components. The main component of 281 samples was eucalyptol, and its relative content in the essential oil was in the range of 11.63–65.33%. A total of 155 samples were mainly composed of linalool, and its relative content in the essential oil was in the range of 11.23–96.45%. A total of 168 and 274 samples were mainly composed of camphor and *trans*-nerolidol, respectively, and their relative contents in the essential oil were in the ranges of 13.89–87.69% and 15.24–88.66%, respectively. Eucalyptol, linalool, camphor, and *trans*-nerolidol with mean relative contents of >10% were considered major compounds. For example, the relative content of eucalyptol in GD-SX-22 essential oil was 65.33%, that of linalool in FJ-YX-29 was 93.73%, that of camphor in HN-CS-14 was 84.35%, and that of *trans*-nerolidol in JX-AF-14 was 60.30% (Table 3). Although the number of monoterpenoids was less than that of sesquiterpenoids, three of the four major compounds (eucalyptol, linalool, and camphor) were monoterpenoids, indicating that monoterpenoids dominated camphor tree essential oil.

Correlation coefficients indicated significant positive or negative correlations among several sets of compounds (Figure 2). Strongly positive correlations, meaning the compounds were present as a group, were observed among *α*-pinene, *β*-phellandrene, eucalyptol, *γ*-terpinene, *trans*-linalool oxide, *β*-terpineol, 4-terpineol, and *α*-terpineol. *β*-caryophyllene, humulene, germacrene D, valencene, *α*-muurolene, germacrene B, selina-6-en-4-ol, and bulnesol formed another positively correlated group. Notably, strongly negative correlations were observed between compounds from the different groups.

Although the mean essential oil compound content ranged from 0.10% to 15.14%, the content of individual compounds varied considerably; a given individual sample often contained one or two main compounds and low amounts of other compounds. The data in Table 3 show that eucalyptol (GD-SX-22), linalool (FJ-YX-29), camphor (HN-CS-14), borneol and camphor (GX-QZ-03), methyl isoeugenol (GX-GF-22), *trans*-nerolidol (JX-AF-14), and selina-6-en-4-ol (GX-GF-06) were the major compounds for their respective individuals. The variety of essential oil components among individual plants reflected the variety of camphor tree chemotypes.

### 2.3. Chemotype Classification and Distribution

Essential oil compounds varied greatly among plants of different chemotypes. Clustering was applied to classify the essential oils of these 974 samples into different chemotypes, depending on the dominant compound in the sample. Five chemotypes were identified, including the eucalyptol type (eucalyptol), nerolidol type (*trans*-nerolidol), camphor type (camphor), linalool type (linalool), and selina type (selina-6-en-4-ol). Briefly, 249 eucalyptol-, 153 camphor-, 140 linalool-, 234 nerolidol-, and 97 selina-type individuals were classified. Furthermore, 84 samples with no predominant compounds were classified as mixed type, and 17 samples that contained other predominant compounds were classified as other type (Table 4).

Four principal components (PCs) accounted for 93.69% of the total variance; therefore, these four PCs were used to explain the variance in essential oil (Table 5). A strongly negative correlation was observed between PC1 and linalool levels. PC2 showed strongly positive correlations with *β*-phellandrene, eucalyptol, and *α*-terpineol and a negative correlation with camphor. PC3 was strongly and positively correlated with *β*-caryophyllene, *trans*-nerolidol, caryophyllene oxide, and 3-methyl-but-2-enoic acid-1,7,7- trimethyl-bicyclo [2.2.1]hept-2-yl ester and negatively correlated with *α*-pinene, *β*-phellandrene, eucalyptol, and *α*-terpineol. PC4 was strongly and positively correlated with valencene, *α*-muurolene, germacrene B, selina-6-en-4-ol, and bulnesol. As shown in Figure 3, PC1 and PC2 accounted for 33.01% and 26.37% of the total variation, respectively, whereas PC3 and PC4 accounted for 24.72% and 9.59% of the total variation, respectively. PC1 was dominant in samples defined as the linalool type; PC2 and PC3 were dominant in samples defined as the camphor, eucalyptol, and nerolidol types; and PC4 was dominant in samples defined as the selina type. Although the four PCs clearly distinguished these chemotypes, various samples did not clearly fit a specific chemotype.

Analysis of variance (ANOVA) revealed significant differences (*p* < 0.01) in essential oil yield among the different chemotypes (Table 4). The yields for the eucalyptol and linalool types (1.64% and 1.55%, respectively) were significantly higher than those for the other five chemotypes (<1.00%), whereas the *trans*-nerolidol chemotype had the lowest yield (0.22%). Within each chemotype, the yields of essential oils showed significant variance. For the eucalyptol type, the highest yield (in the range of 0.09–3.28%) was 36 times higher than the lowest yield; 86 times higher for the camphor type, and 10 times higher for the linalool type. The extreme variability within chemotypes provided sufficient basis for selecting optimal wild individuals.

One or two chemotypes were predominant in each population (Figure 4). The eucalyptol and linalool types were predominant in populations from Qimen, Xianning, Changde, Xinhua, Yongzhou, and Hanzhong; the nerolidol type was predominant in populations from Fuzhou, Shaoguan, Zhenjiang, Tonglu, and Yuyao; the selina type was predominant in populations from Huaan, Yongan, and Youxi; and the camphor type was predominant in populations from Gaofeng and Guiyang.

Pearson’s correlation coefficient analysis of longitude, latitude, and the five major compounds (eucalyptol, linalool, camphor, selina-6-en-4-ol, and *trans*-nerolidol) correlated the variations in essential oil with geographic location (Table 2). A significantly positive correlation was detected between eucalyptol, linalool, and latitude; the higher the latitude, the higher the eucalyptol and linalool contents. Selina-6-en-4-ol was significantly and positively correlated with longitude and significantly and negatively correlated with latitude. *trans*-Nerolidol was positively correlated with longitude, and camphor was negatively correlated with longitude.

### 2.4. Exceptional Chemotypes

Despite the predominance of the eucalyptol, camphor, linalool, nerolidol, and mixed types in camphor trees, 17 individuals were present with high amounts of other compounds (Table 3). Six (GX-GF-22, JX-AF-13, JX-XG-21, SX-HZ-22, and SX-HZ-24, SX-HZ-25) of the 17 individual plants exhibited high amounts of methyl isoeugenol (68.29% to 83.60%). Two individual plants (FJ-FZ-30 and GD-ZJ-04) exhibited high amounts of methyl eugenol (87.71% and 84.19%, respectively). Two individuals (JX-NC-23 and JX-WY-20) had high amounts of *δ*-selinene (74.48% and 60.54%, respectively). One individual (GX-QZ-03) had an elevated borneol content (52.22%).

### 2.5. Impact of Environmental Factors on Essential Oil Variability

In order to estimate the effects of environmental factors on essential oil yield and compound variability, data of four environmental factors were assessed, including mean annual maximum temperature (MAAT), mean annual minimum temperature (MAIT), mean annual precipitation (MAP), and elevation. The environmental factors were inputted as the first group, while oil yield and five major essential oil components (eucalyptol, linalool, camphor, selina-6-en-4-ol, and *trans*-nerolidol) were inputted as the second group for canonical correspondence analysis (CCA) in SPSS 26.0. The four first canonical variables could explain 100% of the total variation (*p*<0.01). The coefficient of the first CCA set explained 55% of the variation and revealed that MAAT positively affected the environmental variation, and camphor negatively affected the essential oil variation. The second canonical set explained 33.2% of the variation and disclosed that MAAT and elevation negatively influenced the coefficient variation in relation to environmental factors, whereas eucalyptol and linalool positively affected the oil yield, while selina-6-en-4-ol was negatively correlated with oil character variation. The third CCA set explained 9% of the variation and had a positive correlation with MAIT and negative correlation with MAAT in relation to environmental factors. The contribution of these coefficients to the third set showed that oil yield, eucalyptol, linalool, and selina-6-en-4-ol had negative effects on oil character variation (Table 6). Based on the first row of CCA sets, high MAAT and low elevation in areas were the major environmental factors influencing the characters of oil yield, eucalyptol, and linalool.

## 3. Discussion

Sect. Camphora (Trew) Meissn spice plants are commercially important and chemically diverse. Almost all *Cinnamomum* plants contain essential oils, as reported in various studies [18,19,20,21]. Dozens of essential oil compounds, including terpenes, terpenoids, and phenylpropanoids, have been identified in *Cinnamomum Trew* plants, enabling categorization by chemotype. The linalool type is chemotypical for *Cinnamomum osmophloeum* [22], *Cinnamomum porrectum* [23], *Cinnamomum kanehirae* [24], *Cinnamomum champhora* [25], and *Cinnamomum verum* [26]; the eugenol type is chemotypical for *Cinnamomum impressinervium* [27] and *C. verum* [28]; and the eucalyptol type is chemotypical for *C. kanehirae* [24] and *C. porrectum* [23]. Some other chemotypes, including the safrole, cinnamaldehyde, citral, camphor, eucalyptol, and nerolidol types, are also found in *Cinnamomum Trew*. [20,23,29,30]. Although the variations in the essential oil composition were minimal for some species, the variation in *Cinnamomum Trew* chemotypes did not depend on differences in species. The 974 samples from 35 populations in 13 provinces of *C. camphora* were divided into six main chemotypes (eucalyptol, camphor, linalool, nerolidol, selina, and mixed types) and four other rare chemotypes (methyl isoeugenol, methyl eugenol, selinene, and borneol types). All the main chemotypes have been reported previously [11,12,13,31], while for the rare chemotypes, only the borneol type has been previously reported in *C. camphora* [11].

Plant essential oils are divided into terpenoids and phenylpropanoids and are synthesized in plants through independent and compartmentally separate pathways. Sesquiterpenes are derived from the terpenoid mevalonate (MVA) pathway, whereas hemiterpenes, monoterpenes, and diterpenes are produced via the methyl-erythritol phosphate (MEP) pathway. Phenylpropanoids are biosynthesized via the shikimate pathway [32]. The monoterpenes eucalyptol, camphor, linalool, and borneol originate from the MEP pathway [33]. The sesquiterpenes *trans-*nerolidol, selina-6-en-4-ol, and δ-selinene are derived from the MVA pathway [32]. The shikimate pathway produces methyl isoeugenol/methyl eugenol [34]. All three biosynthetic pathways are present in the camphor tree, wherein they regulate the synthesis of essential oils depending on genotype, developmental stage, and biotic stressors.

Positively correlated compounds may be synthesized by the same enzyme or by different products in the same synthetic pathway in the camphor tree, whereas negatively correlated chemicals may have a substrate competition relationship or an upstream and downstream relationship in the same synthetic route. *α*-Terpineol has been proposed to be the precursor of eucalyptol; enzymes catalyze protonation and internal addition of the endocyclic double bond of *α*-terpineol to produce eucalyptol. Thus, *α*-terpineol, *β*-pinene, *α*-pinene, sabinene, and myrcene are minor products of the conversion of (2E)-geranyl diphosphate to eucalyptol [35]. In this study, a significantly positive correlation was detected between eucalyptol, *α*-terpineol, and *α*-pinene levels. Furthermore, camphor is converted from borneol by the NAD-dependent dehydrogenation of the alcohol to a ketone [36,37,38]. Borneol dehydrogenase is the key enzyme that converts borneol into camphor, and its activity plays a decisive role in essential oil camphor content. Two individual borneol-type camphor tree plants have previously been found in Jiangxi Province [11]. In the current study, one borneol type (GX-QZ-03, 52.22%) and four mixed types with high borneol content (GX-QZ-08, GX-QZ-05, SC-LX-03, and SC-LX-23; 46.61%, 36.58%, 37.71%, and 42.17%, respectively) and high camphor content (23.37%, 31.48%, 41.42%, 39.51%, and 34.35%, respectively) were found. This finding suggests that borneol-type camphor trees are rare. Borneol was present in the essential oil and was closely correlated with camphor, suggesting that the borneol type originates from a decrease in borneol dehydrogenase activity; borneol-type samples may be promising materials for studying borneol dehydrogenases.

Although camphor trees are native to subtropical monsoon climates, their essential oil contents varied according to a geographical gradient. Northern camphor trees had higher eucalyptol and linalool contents and lower selina-6-en-4-ol content. Eastern trees had higher selina-6-en-4-ol and *trans*-nerolidol contents and lower camphor content. These trends show that the biosynthesis and accumulation of essential oils may be affected by environmental factors, as well as genetic mutations. Rainfall and temperature are the most important environmental factors associated with various essential oil yields and compositions [39,40]. China is a suitable area for camphor trees since the annual mean temperature decreases from south to north, ranging from 23 °C (Zhanjing) to 14 °C (Hanzhong). The annual rainfall decreases from east to west, ranging from 1800 mm (Yuyao) to 750 mm (Luxian). Thus, the biosynthesis and accumulation of essential oils by camphor trees may be affected by rainfall and temperature at their locations. Individuals rich in eucalyptol and linalool inhabit regions with low annual mean temperatures. Plants rich in selina-6-en-4-ol are found in Fujian and Guangdong provinces of southeast China, which have a higher annual mean temperature and higher annual mean rainfall. Furthermore, the plant organ developmental stage can influence biosynthesis and accumulation of essential oils; thus, the yields and profiles of essential oils change seasonally. The yield of essential oil and the proportion of linalool in the essential oil in linalool-type camphor trees were highest during the rainy season from May to July [41]. To minimize the effect of organ development stage on essential oil yield and profile, we collected plant samples from May to September for this study.

Camphor trees have been cultivated as a crop for many decades in China to produce camphor, linalool, eucalyptol, and borneol, but we also found individual plants rich in methyl isoeugenol, methyl eugenol, or δ-selinene. Methyl isoeugenol has been reported in *Cymbopogon nardus* and *Pimenta pseudocaryophyllus* [42,43], while methyl eugenol has been found in *Croton malambo* [44], and δ-selinene has been reported in *Jatropha elliptica* rhizomes [45]. Although these chemotypes rarely occur in camphor trees, they are valuable for developing new essential oil products.

Camphor trees are wildly distributed and they can adapt to different environments; therefore, it is necessary to determine the influence of environmental factors on essential oil heterogeneity. The results of CCA showed a strong relationship between environmental factors and essential oil constituents, indicating that the essential oils of camphor trees are greatly affected by the environment, similar to the results reported in other studies.

## 4. Materials and Methods

### 4.1. Plant Material

Leaves of more than 10 individual plants from each population were collected and the weights were measured immediately. Then, the samples were separately packed into valve bags to take back to the laboratory for analysis. A total of 974 individual plants from 35 populations in 13 provinces of China were sampled from May to September 2019 (Table 1). The geographic location (latitude and longitude) of each plant was recorded, and one herbarium specimen was collected and taxonomically identified according to the description of the flora in China [2].

### 4.2. Extraction of the Essential Oils

The essential oil was obtained from each of the samples using a hydrodistillation method. The extracts of fresh leaves (~100 g) were distilled (3 h) using a modified Clevenger-type apparatus. After 3 h, the oil was collected with a centrifugal tube and approximately 1 g anhydrous sodium sulfate was added into the tube. Then, the oil was poured into another tube after shaking for approximately 10 s and weighed. Solutions of essential oil (3%) in ethanol were analyzed using gas chromatography-mass spectrometry (GC-MS) and gas chromatography-flame ionization detection (GC-FID).

### 4.3. Essential Oil Compound Identification

GC-MS for compound identification was performed using a SHIMADZU GC-2010 Plus apparatus (Shimadzu corporation, Japan) equipped with a GCMS-QP2020 mass spectrometer coupled with an SH-Rxi-5Sil MS quartz capillary column (30 m × 0.25 mm, film thickness 0.25 μm). Helium (1.0 mL/min) was used as the carrier gas. Samples (1 μL; split ratio, 1:20) were injected into the GC (detector temperature, 280 °C), and subjected to the following temperature ramping program: initial temperature (60 °C; 2 min), followed by ramping (5 °C/min) to the final temperature (220 °C; 20 min). Qualitative analysis used electron-impact ionization (source temperature, 200 °C; interface temperature, 250 °C; ionization energy, 70 eV; scanning speed, 1000 mm/s; scanning interval, 0.50 fragments per second; mass scan range, 40–650 amu). Essential oils were identified using a digital library of mass spectral data (NIST 8.0) and literature comparisons of retention indices calibrated against a homologous series of C8-C32 *n*-alkanes [46].

### 4.4. Compound Calculation

GC-FID quantitative analysis was performed using a SHIMADZU GC-2010 Plus instrument (Shimadzu corporation, Japan) fitted with an SH-Rxi-5Sil MS quartz capillary column under the same conditions as GC-MS, except that N_2_ was used as the carrier gas. The FID detector temperature was 250 °C. The relative percentage of each compound in the essential oil was estimated using the peak area normalization method, based on the ratio of the area of the respective peak to the total area of all compounds in the sample; compound response factors were considered equivalent.

### 4.5. Statistical Analysis

The 41 compounds identified in the essential oil were analyzed collectively. ANOVA was performed between the chemotypes and frequency distributions. Pearson’s correlation coefficients of essential oil yields and latitude or longitude and CCA analysis were calculated using IBM SPSS Statistics software version 26.0 (IBM Corp., Armonk, NY, USA). Principal component analysis and correlation coefficient analysis between the essential oil compounds were conducted using the factoextra and corrplot packages in R, respectively (The R Foundation for Statistical Computing, Vienna, Austria).

## 5. Conclusions

*C. camphora* in China exhibited diversity in essential oil yield and profile. The Qimen, Xinhua, Hanzhong, and Enshi populations had high oil yields, with mean yields of >1.5%. Leaf essential oils contained 41 identifiable compounds and were classifiable into six major chemotypes and several other rare chemotypes. Among them, five major chemotypes were classified: eucalyptol, camphor, linalool, nerolidol, and selina types. Individuals with no predominant constituent were classified as the mixed type.

Geographic temperature and rainfall gradients could rationalize trends in essential oil yield and composition. With increasing latitude, oil yield and eucalyptol and linalool contents increased, whereas selina-6-en-4-ol content decreased. In contrast, selina-6-en-4-ol and *trans*-nerolidol contents increased with increasing longitude, whereas yield and camphor content decreased. Significant differences in oil yield were observed among chemotypes, with the eucalyptol type having the highest yields, followed by the linalool, camphor, nerolidol, selina, and mixed types in order of decreasing oil yield. Despite one or two chemotypes having been commercialized and several clones of camphor trees available to extract essential oils, other chemotypes or high-yield individuals could be applied as new, valuable commercial resources. Additionally, the geographical distribution of essential oils provides a valuable reference for the regional allocation of agricultural production for these oils.

## Figures and Tables

**Figure 1 molecules-28-00973-f001:**
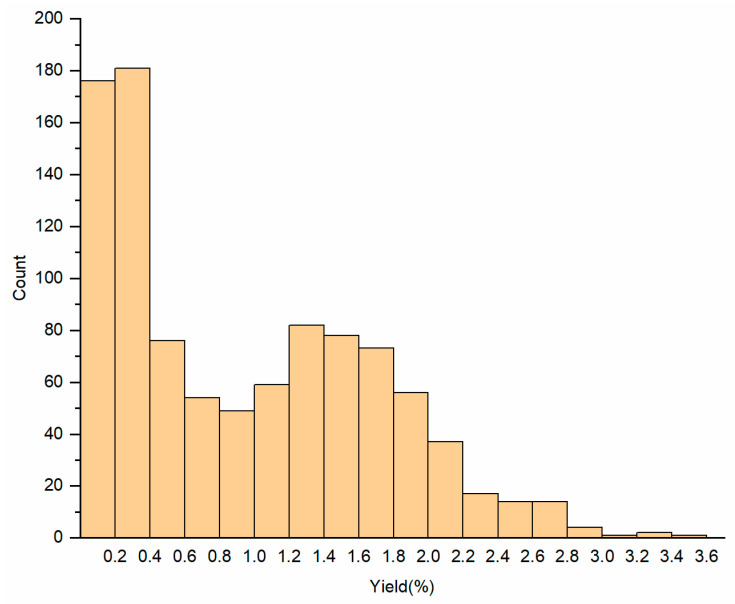
Frequency distribution of the essential oil yield of *C. camphora*.

**Figure 2 molecules-28-00973-f002:**
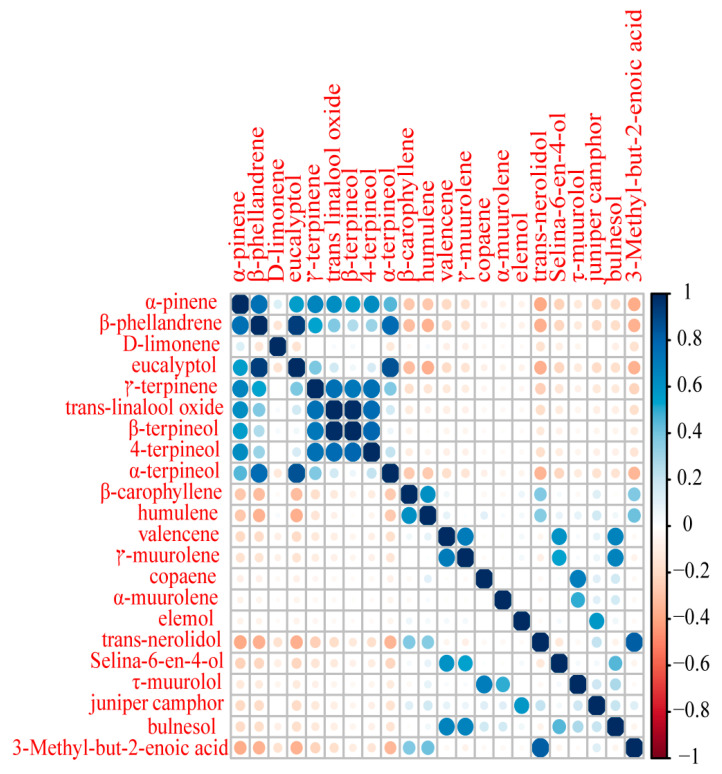
Correlation analysis of the 22 compounds identified in the essential oil of *C. camphora*.

**Figure 3 molecules-28-00973-f003:**
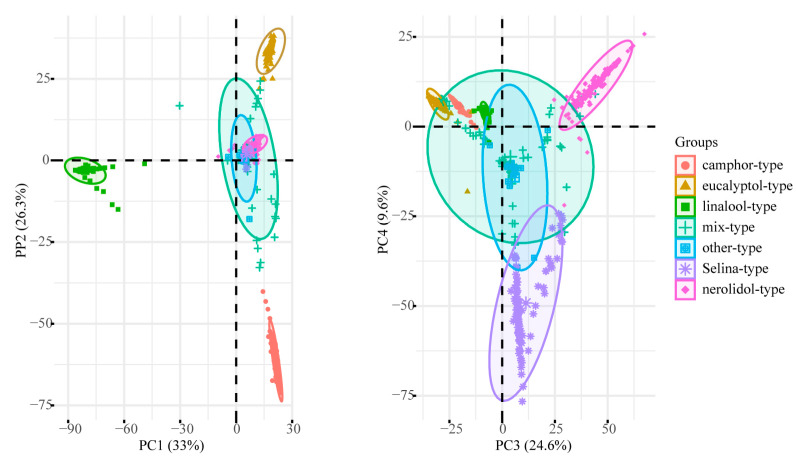
Principal component analysis of the first four dimensions.

**Figure 4 molecules-28-00973-f004:**
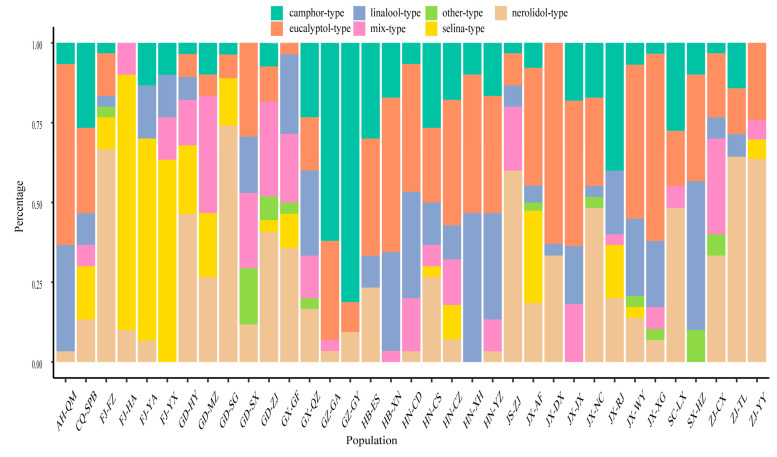
Distribution frequencies of chemotypes in different populations.

**Table 1 molecules-28-00973-t001:** Geographical information of samples of *C. camphora* populations with mean essential oil yield of each population.

Site	Province	Code	N ^1^	E	MAAT	MAIT	MAP	Elevation	n	Oil Yield (%)
Qimen	Anhui	AH-QM	29.696716°	117.506601°	23	13	2396	90	30	2.01 ± 0.74
Shapingba	Chongqing	CQ-SPB	29.579210°	106.417387°	23	16	982	465	30	0.92 ± 0.16
Fuzhou	Fujian	FJ-FZ	26.157219°	119.283930°	27	18	1836	86	30	0.49 ± 0.50
Huaan	Fujian	FJ-HA	24.959486°	117.534795°	28	17	2989	205	10	0.48 ± 0.14
Yongan	Fujian	FJ-YA	26.011571°	117.390753°	27	16	2876	224	30	0.61 ± 0.44
Youxi	Fujian	FJ-YX	26.040294°	118.128517°	27	16	2769	176	30	0.55 ± 0.35
Heyuan	Guangdong	GD-HY	23.718087°	115.209347°	28	17	2648	343	28	0.66 ± 0.63
Meizhou	Guangdong	GD-MZ	24.146901°	116.076344°	28	18	2099	132	30	0.68 ± 0.59
Shaoguan	Guangdong	GD-SG	24.849486°	113.562527°	26	17	2814	82	27	0.28 ± 0.25
Shixing	Guangdong	GD-SX	24.855446°	114.100072°	27	16	2867	199	17	0.95 ± 0.92
Zhanjiang	Guangdong	GD-ZJ	21.126940°	110.246939°	28	21	1909	13	27	0.36 ± 0.47
Gaofeng	Guangxi	GX-GF	22.931939°	108.357974°	27	19	2014	118	28	1.10 ± 0.89
Quanzhou	Guangxi	GX-QZ	25.986144°	110.921399°	25	16	1979	184	30	0.91 ± 0.68
Guian	Guizhou	GZ-GA	26.412591°	106.369501°	20	13	1490	1226	29	1.47 ± 0.67
Guiyang		GZ-GY	26.496424°	106.735431°	19	12	1981	1124	32	1.00 ± 0.66
Enshi	Hubei	HB-ES	30.281372°	109.484778°	23	14	1263	430	30	1.51 ± 0.86
Xianning	Hubei	HB-XN	29.569482°	114.862824°	23	13	1445	57	29	1.41 ± 0.46
Changde	Hunan	HN-CD	29.050278°	111.592693°	22	15	1327	91	30	1.35 ± 0.45
Changsha	Hunan	HN-CS	28.111381°	113.054928°	24	16	1591	90	30	0.69 ± 0.57
Chenzhou	Hunan	HN-CZ	25.789165°	112.996943°	23	15	2524	211	28	1.09 ± 0.70
Xinhua	Hunan	HN-XH	27.780197°	111.180908°	24	15	2204	181	30	1.70 ± 0.46
Yongzhou	Hunan	HN-YZ	26.391109°	111.761389°	24	16	822	176	30	1.02 ± 0.49
Zhenjiang	Jiangsu	JS-ZJ	32.175807°	119.441861°	21	12	1444	33	30	0.41 ± 0.47
Anfu	Jiangxi	JX-AF	27.406026°	114.560550°	25	15	2656	116	38	1.18 ± 1.00
Dexing	Jiangxi	JX-DX	28.844566°	117.579117°	25	15	2478	49	27	1.09 ± 0.84
Jinxi	Jiangxi	JX-JX	27.871332°	116.584137°	25	15	2821	47	11	0.89 ± 0.55
Nanchang	Jiangxi	JX-NC	28.779930°	115.779720°	24	16	2515	129	29	0.78 ± 0.69
Ruijin	Jiangxi	JX-RJ	25.864169°	115.932219°	26	16	2604	223	30	0.80 ± 0.54
Wuyuan	Jiangxi	JX-WY	29.252720°	117.859167°	25	14	2787	80	29	1.12 ± 0.73
Xingguo	Jiangxi	JX-XG	26.495143°	115.771943°	26	17	2341	265	29	1.42 ± 0.68
Luxian	Sichuan	SC-LX	29.138419°	105.385134°	23	15	1168	478	29	0.47 ± 0.35
Hanzhong	Shaanxi	SX-HZ	33.135874°	107.301057°	22	12	1764	443	30	1.54 ± 0.49
Changxing	Zhejiang	ZJ-CX	30.977313°	119.609480°	23	14	1614	115	30	0.60 ± 0.62
Tonglu	Zhejiang	ZJ-TL	29.818942°	119.757321°	23	13	2392	51	14	0.77 ± 0.65
Yuyao	Zhejiang	ZJ-YY	30.038909°	120.992886°	23	14	2972	19	33	0.44 ± 0.44

^1^ Abbreviations: N: latitude, E: longitude, MAAT: mean annual maximum temperature, MAIT: mean annual minimum temperature, MAP: mean annual precipitation.

**Table 2 molecules-28-00973-t002:** Pearson correlation coefficients between essential oil and geographical position.

	Latitude	Longitude
Yield	0.162 **	−0.174 **
Eucalyptol	0.175 **	−0.014
Linalool	0.096 **	−0.059
Camphor	−0.014	−0.282 **
Selina-6-en-4-ol	−0.224 **	0.154 **
*trans*-Nerolidol	0.003	0.189 **

** Correlation was significant at the level of 0.01.

**Table 3 molecules-28-00973-t003:** Relative content of 41 compounds in essential oil and samples with different chemotypes.

Constituents	RI ^1^	Mean Content ^2^ (%)	GD-SX -22	FJ-YX -29	HN-CS -14	JX-AF -14	GX-QZ -03	GX-GF -22	GX-GF -06
*α*-Pinene	939	1.20	2.91	-	0.63	0.06	1.56	-	-
*β*-Phellandrene	978	3.75	10.47	-	-	0.04	0.35	0.04	-
*α*-Phellandrene	1011	0.10	0.01	-	-	0.04	0.32	0.06	-
D-limonene	1035	0.40	-	-	1.15	0.17	1.90	0.05	-
Eucalyptol	1039	15.14	65.33	0.03	1.63	0.38	1.77	0.07	-
*γ*-Terpinene	1063	0.26	-	-	-	-	-	0.17	-
*trans*-Linalool oxide	1074	0.54	0.01	0.15	0.22	0.03	0.42	3.78	-
Linalool	1106	13.57	1.59	93.73	-	0.39	2.87	0.44	0.02
*β*-Terpineol	1109	0.28	-	-	-	0.04	0.16	3.58	-
Camphor	1155	12.77	-	0.23	84.35	0.75	23.37	0.01	0.02
Borneol	1181	0.47	0.13	-	0.22	0.09	52.22	-	0.15
4-Terpineol	1188	0.95	2.99	0.04	0.52	0.03	0.46	4.49	0.02
*α*-Terpineol	1205	3.75	9.25	0.27	1.03	0.23	1.32	0.43	0.15
*trans*-Geraniol	1229	0.13	0.20	-	-	0.04	-	0.02	-
Bornyl acetate	1290	0.14	-	-	-	0.14	-	0.13	0.03
*δ*-Elemene	1342	0.43	0.05	0.08	0.30	0.17	0.58	0.36	0.53
Methyl eugenol	1401	0.21	-	0.01	-	-	-	7.03	-
*β*-Caryophyllene	1433	2.93	0.54	0.37	2.32	2.92	1.91	3.05	4.35
*γ*-Elemene	1439	0.10	0.01	-	-	0.09	-	-	0.01
Humulene	1469	1.53	0.01	-	1.14	0.89	1.28	2.36	2.46
Valencene	1485	0.23	0.01	-	-	0.03	-	0.07	0.62
γ-Muurolene	1483	0.20	-	-	-	-	-	0.02	1.00
Germacrene D	1489	1.04	0.17	0.31	1.40	0.93	0.41	-	0.34
*δ*-Selinene	1496	0.88	-	-	1.22	-	1.87	-	7.80
*α*-Guaiene	1503	0.89	0.06	0.07	1.27	0.04	1.33	-	0.29
Methyl isoeugenol	1509	1.73	0.01	0.43	-	0.32	-	71.99	0.30
Copaene	1542	0.36	-	-	-	0.03	-	0.01	0.07
*α*-Muurolene	1547	0.25	-	-	-	-	-	0.01	-
Elemol	1559	0.29	-	0.02	0.23	0.68	-	0.05	0.52
*trans*-Nerolidol	1564	13.93	-	0.02	0.23	60.30	-	0.04	0.43
Germacrene B	1574	0.50	0.05	0.07	-	0.01	-	0.02	1.13
Spathulenol	1589	0.85	0.01	0.10	-	0.46	-	0.02	1.90
caryophyllene oxide	1598	0.90	0.03	0.05	-	1.03	-	0.03	0.53
Guaiol	1609	0.24	0.04	-	-	2.71	-	0.12	-
Selina-6-en-4-ol	1637	5.58	-	0.02	-	0.11	-	0.01	71.20
Viridiflorol	1648	0.10	-	-	-	0.07	-	0.02	0.28
*τ*-Muurolol	1657	0.34	0.01	-	-	0.06	-	-	0.20
*α*-Cadinol	1667	0.18	-	-	-	-	0.17	-	0.81
Juniper camphor	1672	0.52	0.07	0.01	-	0.78	-	0.13	-
Bulnesol	1678	0.28	0.01	0.03	-	1.21	-	0.06	0.41
3-Methyl-but-2-enoic acid-1,7,7- trimethyl-bicyclo [2.2.1]hept-2-yl ester	1731	4.39	0.46	-	-	20.88	-	0.03	0.36
Monoterpene hydrocarbons		19.28	13.39	-	1.78	0.31	4.13	0.32	-
Oxygenated Monoterpenes		34.17	79.50	94.45	87.97	2.12	82.59	12.95	0.39
Sesquiterpenes hydrocarbons		9.34	0.85	0.90	7.65	5.11	7.38	5.90	18.60
Oxygenated sesquiterpenes		27.60	0.68	0.25	0.46	88.29	0.17	0.51	76.64
Phenylpropanoids		1.94	0.01	0.44	-	0.32	-	79.02	0.30

^1^ Retention index (RI) relative to the homologous series of n-hexane on the SH-Rxi-5Sil MS quartz capillary column. ^2^ Average relative content of a constituent in all sample essential oils.

**Table 4 molecules-28-00973-t004:** The quantity and oil yield of each chemotype.

Chemotype	n	Mean Yield (%)	Min. Yield (%)	Max. Yield (%)
Eucalyptol	249	1.64 ± 0.04a ^1^	0.09	3.28
Camphor	153	0.95 ± 0.04b	0.04	3.46
Linalool	140	1.55 ± 0.04a	0.27	2.83
*trans*-Nerolidol	234	0.22 ± 0.01e	0.01	1.41
Selina-6-en-4-ol	97	0.45 ± 0.02d	0.12	2.07
Mix	84	0.66 ± 0.07c	0.05	2.76
Other	17	0.34 ± 0.08de	0.01	0.99

^1^ Different letter in the same column represented significant differences (*p* < 0.01).

**Table 5 molecules-28-00973-t005:** Four principal components of the 41 essential oil compounds of *C. camphora*.

Compounds	PC1	PC2	PC3	PC4
*α*-Pinene	0.36	0.18	−0.54	−0.14
*β*-Phellandrene	0.32	0.62	−0.62	−0.19
*α*-Phellandrene	0.034	−0.06	−0.04	0.01
D-limonene	0.13	−0.45	−0.15	−0.04
Eucalyptol	0.33	0.65	−0.64	−0.21
*γ*-Terpinene	0.18	0.26	−0.32	−0.05
*trans*-Linalool oxide	0.08	0.13	−0.19	−0.00
Linalool	−0.99	−0.04	−0.11	−0.10
*β*-Terpineol	0.06	0.09	−0.09	0.04
Camphor	0.28	−0.89	−0.31	−0.14
Borneol	0.06	−0.13	−0.04	−0.01
*4*-Terpineol	0.10	0.14	−0.16	0.02
*α*-Terpineol	0.30	0.56	−0.58	−0.18
*trans*-Geraniol	0.03	0.04	0.02	0.01
Bornyl acetate	0.11	−0.27	−0.05	−0.05
*δ*-Elemene	0.14	−0.10	0.36	0.11
Methyl eugenol	0.01	0.01	0.01	0.04
*β*-caryophyllene	0.15	−0.10	0.56	−0.00
*γ*-Elemene	0.04	0.02	0.18	0.08
Humulene	0.19	−0.14	0.46	0.08
Valencene	0.05	0.02	0.15	0.61
γ-Muurolene	0.04	0.02	0.09	0.55
Germacrene D	0.18	−0.20	0.46	−0.20
*δ*-Selinene	0.05	−0.06	0.06	0.40
*α*-Guaiene	0.21	−0.42	0.10	−0.03
Methyl isoeugenol	0.03	0.05	0.17	0.02
Copaene	0.02	0.01	0.04	0.10
*α*-Muurolene	0.01	0.01	0.02	0.06
Elemol	0.01	−0.01	0.03	0.09
*trans*-Nerolidol	0.18	0.10	0.91	−0.33
Germacrene B	0.04	0.04	0.15	0.60
Spathulenol	0.07	0.03	0.30	0.39
caryophyllene oxide	0.11	0.05	0.53	0.20
Guaiol	0.05	0.04	0.19	0.12
Selina-6-en-4-ol	0.06	0.02	0.14	0.95
Viridiflorol	0.04	0.02	0.18	0.11
*τ*-Muurolol	0.03	0.01	0.07	0.15
*α*-Cadinol	0.04	0.00	0.12	0.26
Juniper camphor	0.07	0.03	0.29	0.09
Bulnesol	0.05	0.02	0.12	0.52
3-Methyl-but-2-enoic acid-1,7,7- trimethyl-bicyclo [2.2.1]hept-2-yl ester	0.16	0.10	0.81	−0.21
% of Variance	33.01	26.37	24.72	9.59

**Table 6 molecules-28-00973-t006:** Canonical coefficients for four CCA sets between phytochemicals with environmental factors.

Characters		Standardized Canonical Coefficients			
		1	2	3	4
Environmental factors	MAAT	0.94	−1.05	−1.55	−1.41
	MAIT	−0.26	0.34	1.95	0.68
	MAP	0.06	−0.19	0.31	1.25
	Elevation	−0.35	−1.12	0.00	−0.23
Oil characters	Oil yield	−0.19	−0.60	−0.59	0.16
	Eucalyptol	−0.46	0.75	−0.58	1.21
	Linalool	−0.25	0.58	−0.74	0.30
	Camphor	−0.99	−0.25	−0.21	0.77
	*trans*-Nerolidol	−0.33	0.22	−0.35	1.42
	Selina-6-en-4-ol	0.25	−0.57	−0.64	0.86
Eigen-value		0.46	0.38	0.21	0.12
*p* value		0.00	0.00	0.000	0.00
Cumulative variance		55.00	88.20	97.20	100.00

## Data Availability

All the data are shown in the main manuscript and in the Appendix A.

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
