# Peer review of "Chemical Variation and Environmental Influence on Essential Oil of Cinnamomum camphora"

_molecules, 2023, doi:10.3390/molecules28030973_

Round 1

Reviewer 1 Report

The authors report geographic relationships for the yields and compositional diversity of C. camphora essential oils. The essential oils of 974 individual trees from 35 populations in 13 provinces were extracted by hydrodistillation and analyzed qualitatively and quantitatively by gas chromatography-mass spectrometry and gas chromatography-flame ionization detection, respectively. The article generally gave a good background introduction and objectives, as well as sound experimental techniques. It can be published in this journal after the following minor revisions:

If you talk about the impact of environmental factors on the variability of the essential oils obtained, it is important to incorporate, in M&M, numerical information (mean +/- SD) of the mean-annual maximum temperature (MAAT), mean-annual minimum temperature (MAIT), and mean annual precipitation (MAP) of the 35 populations in the 13 provinces of China in which the plants were collected.

3. Discussion:

Lines 251 and 252: Please, the first time, write the full name and the acronym in parentheses: MVA and MEP, respectively.

Author Response

  1. We have added numerical information in Table 1 according to the reviewer.
  2. We have writen the full name of MVA and MEP appeared for the first time.

Reviewer 2 Report

The essential oils from Cinnamomum Trew plants, including C. camphora, are very important because they are attributed several biological properties, such as antibacterial, antifungal, antiviral, anti-inflammatory, anti-proliferative, anxiolytic, skin regenerating, among others. The knowledge of the variability of this composition is important to ensure that the product has quality and is safe for the consumer. In this MS, Zhang et al presents a laborious work where the yields and compounds diversity of C. camphora essential oil of 974 individuals from 35 populations in Southern China, and establishes a relationship between the essential oil with environmental factors. The MS is well written and presents good background literature research. The conclusions were derived from the obtained results during the performed studies and concluded on the basis of the carefully selected scientific literature data. I suggest some compulsory revisions to improve the overall quality of the MS.   Minor comments:

Line 14, C. camphora should be italic.

Line 50, 10 thousand?

Lines 56-58: please insert references supporting this paragraph.

Lines 66-70, Most C. camphora outside China, including Madagascar, Australia etc., are introduced from other countries, and there is no positive connection between the chemotype and the location. This paragraph has nothing to do with this article, it is suggested to delete it.

Line 83, Section 4?

Lines 90-92: The meaning of this sentence is unclear and needs to be revised.

Line 104, Table1 Each population contains multiple individuals, so the latitude and longitude should be in a range, and the oil yield should be the mean plus the standard deviation.

Line 125, What is the mean of “Mean content” in table 3?

Lines 312-313, Please rewrite this sentence, it is not clear.

Line 321, The method should be specified, such as the concentration of anhydrous sodium sulfate, the brand and origin of the instrument used etc.

Author Response

  1. Line 14, camphora had been revised.
  2. In China, the area of camphor tree was planted as spice raw material forest distributed mainly in Jiangxi, Guangxi and Fujian province. Some unofficial evidence shows that the planting area of Jiangxi is 10,000 ha., Fujian is 2,000 ha., Guangxi is 500 ha.. To be conservative, we wrote as "more than10 thousand".
  3. We have inserted references supporting in the corresponding position.
  4. We have deleted the paragraph according to the reviewer suggestion.
  5. To avoid ambiguous expression, the sentence has been revised.
  6. The sentence has been revised to “Additionally, several individuals with particularly high essential oil content of > 3.0%, including HB-ES10 (3.46%), AH-QM20 (3.28%), JX-AF11 (3.24%) and JX-AF38 (3.03%), were potentially excellent individuals for commercial development.”
  7. We have added the standard deviation data in the table.
  8. We have added a footer under the tabel to explain the mean of “Mean content”.
  9. The sentence has been revised to “Leaves of more than 10 individual plants from each population were collected and the weights were measured immediately. Then the sample was packed into a valve bag separately to take back to the laboratory for analysis. In total 974 individual plants from 35 populations in 13 provinces of China were collected from May to September 2019 (Table 1).”
  10. We have added the origin of the instrument in the section 4 according to the reviewer.

Reviewer 3 Report

This study is interesting, as the Cinnamomum trees are important source of biologically active metabolites. However, some comments are listed below.

Authors analyzed essential oil composition diversity of Camphor trees (Cinnamomum camphora) 35 populations in 13 provinces of China (sampling leaves from 974 individual trees). Lines 312-314; it is not clearly stated how sampling was made (“leaves of more than 10 individual plants from each population or 974 individual plants …” – not clear).

 Lines 321-322; “After 3 h, the oil was dried with anhydrous sodium sulfate and weighed.” – please specify how it was made – was anhydrous sodium sulfate added into the essential oil or some other procedure was applied – please add detailed information.

Authors not mentioned about safrole, frequently present in the cinnamomum oil. We have only one mention about safrole in citation of other research (see line 52). The multistage procedures are usually applied to obtain camphor from the Cinnamon bark or  leaf oil,  to eliminate safrole which is carcinogenic. Please explain, was safrole detected in the described experiments, and, if not, why? 

The Cinnamomum bark provenience is some cases confusing because of the complicated taxonomy. Recently (2022) the work about taxonomy was released – the molecular phylogenetic study that revised classification of the genus Camphora (previously in some cases considered to be the same as Cinnamomum) – see line 232 – it is not clear (what means; “Sect. Camphora (Trew) Meissn spice plants” – not clear – which taxonomic system was used). The work by Zhi Y et al. from Ecology and Evolution (2022), 12 (10) entitled “Phylogeny and taxonomy of Cinnamomum (Lauraceae)” should be take under consideration when discussing the results (section 3 Discussion).

Figure 1 is not very informative – additional explanation needed and reconsidering of the title of the caption.

In the Table 1 (page 3 and 4), please insert the additional column entitled “Province” – so I this case the first column would be “Site”, the second “Province” the third “Code” etc.  In this case we have; e.g. Enshi, Hubei, HB-ES, not Enshi, Hubei province, HB-ES – it will be more clearly presented to the readers avoid multiplication of the word “province” in the whole Table 1.

In the Figure 2 it should be as in the Table 3 ; ß-terpineol (not ß-terpinol) and the same with 4-terpineol and α-terpineol (please unify).

Line 186; should be “…nerolidol type…”

Line 226; should be “…first two CCA sets…” (?)

Conclusions – lines 367-369; “a few clones of camphor trees …” (?) 

Author Response

  1. The sentence has been revised as “Leaves of more than 10 individual plants from each population were collected and the weights were measured immediately. Then the sample was packed into a valve bag separately to take back to the laboratory for analysis. In total 974 individual plants from 35 populations in 13 provinces of China were collected from May to September 2019 (Table 1).” to describe how to make the samples.
  2. We have added the sectences to describe the procedure of oil dried with anhydrous sodium sulfate. “After 3 h, the oil was collected with a centrifugal tube and added about 1 g anhydrous sodium sulfate into the tube. Then the oil was poured into another tube after shaking for about 10 seconds and weighed.”
  3. Safrole is a common compound in the root of Cinnamomum Schaeff plant, also it existes in some Cinnamomum Schaeff plant leaf oil, such as Cinnamomum pauciflorum Nees, but it rarely exists in Cinnamomum camphora. We did not found safrole in the 974 samples.
  4. We referred to the traditional classification method, please see http://www.iplant.cn/info/Sect.%20Camphora?t=z. Sect. Camphora (Trew) Meissn is subordinate taxonomy of Cinnamomum Schaeff. We are concerned aboutessential oil of in this paper, so we think it is not appropriate to discuss classification of Cinnamomum Schaeff.
  5. Additional explanation about Figure 1 has been added in section 2.1 and the tilte of the caption has been revised.
  6. We have been revised Table 1 according to the reviewer.
  7. The misspelling of Figure 2 has been revised.
  8. “nerolido type” has been revised as “nerolidol type”.
  9. The first two CCA sets could explain 88.2% of the variance, so the first two CCA sets should reflect the relation of envirenment and oil characters.
  10. We have revised “a few” as “several” to express more exactly.